# Host and fungal factors both contribute to cryptococcosis-associated hyperammonemia (cryptammonia)

Rosanna P. Baker,[1] Maria Schachter,[2] Steven Phillips,[3] Sheetal Kandiah,[4] Mirza Farrque,[5] Arturo Casadevall,[1] Prem A. Kandiah[2]

**ABSTRACT** *Cryptococcus neoformans* and *Cryptococcus gattii* are both known urease producers and have the potential to cause hyperammonemia. We hypothesized that the risk of hyperammonemia is increased by renal failure, burden of cryptococcal infection, and fungal strain characteristics. We performed a retrospective review of plasma ammonia levels in patients with cryptococcal infections. Risk factors for hyperammonemia were statistically compared between patients with and without hyperammonemia (>53 µmol/L). Cryptococcal cells from three patients included in the study were recovered from our biorepository. Strain characteristics including urease activity, ammonia production, growth curves, microscopy, melanin production, and M13 molecular typing were analyzed and compared with a wild-type (WT) *C. neoformans* strain. We included 29 patients, of whom 37.9% had hyperammonemia, 59% had disseminated cryptococcal infection (DCI), and 41% had isolated central nervous system infection. Thirty-eight percent of patients had renal failure and 28% had liver disease. Renal failure was associated with 4.4 times (95% confidence interval [CI] 1.5, 13.0) higher risk of hyperammonemia. This risk was higher in DCIs (RR 6.2, 95% CI 1.0, 40.2) versus isolated cryptococcal meningitis (RR 2.5, 95% CI, 0.40, 16.0). Liver disease and cryptococcal titers were not associated with hyperammonemia. *C. neoformans* from one patient with extreme hyperammonemia demonstrated a 4- to 5-fold increase in extracellular urease activity, slow growth, enlarged cell size phenotypes, and diminished virulence factors. Hyperammonemia was strongly associated with renal failure in individuals with DCI, surpassing associations with liver failure or cryptococcal titers. However, profound hyperammonemia in one patient was attributable to high levels of urease secretion unique to that cryptococcal strain. Prospective studies are crucial to exploring the significance of this association.

**IMPORTANCE** *Cryptococcus* produces and secretes the urease enzyme to facilitate its colonization of the host. Urease breaks down urea into ammonia, overwhelming the liver's detoxification process and leading to hyperammonemia in some hosts. This underrecognized complication exacerbates organ dysfunction alongside the infection. Our study investigated this intricate relationship, uncovering a strong association between the development of hyperammonemia and renal failure in patients with cryptococcal infections, particularly those with disseminated infections. We also explore mechanisms underlying increased urease activity, specifically in strains associated with extreme hyperammonemia. Our discoveries provide a foundation for advancing research into cryptococcal metabolism and identifying therapeutic targets to enhance patient outcomes.

**KEYWORDS** *Cryptococcus neoformans*, *Cryptococcus*, hyperammonemia, urease, acetohydroxamic acid, cryptammonia, ammonia, mycology

Address correspondence to Prem A. Kandiah, prem.kandiah@emoryhealthcare.org.

Rosanna P. Baker and Maria Schacter contributed equally to this article. Author order was determined alphabetically.

The authors declare no conflict of interest.

Pathogenic microbes can damage organs directly or through the products of their metabolism. Urease-producing microbes that hydrolyze urea to ammonia can cause hyperammonemia, which can result in life-threatening neurological injury. *Cryptococcus neoformans and Cryptococcus gattii* are both recognized as urease producers; however, it remains unclear whether clinical infections are associated with hyperammonemia.

We recently reported a case of profound and clinically significant cryptococcosis-associated hyperammonemia and found two other cases reported in the medical literature (1–3). All three cases were accompanied with renal failure and a disseminated infection. To better clarify the frequency and clinical significance of hyperammonemia in cryptococcosis, we devised a two-part study to explore this question further. During part one of the study, we aimed to retrospectively evaluate the incidence and severity of hyperammonemia in patients with cryptococcal infections. We hypothesized that the occurrence of renal failure and disseminated infection would be a risk factor for hyperammonemia. In part two of the study, we performed a microbiological analysis of available isolates stored in our infectious disease repository that were isolated from the patients included in part one. We hypothesized that strain characteristics could be responsible for increased ammonia production due to increased urease activity. We also evaluated if the rate of ammonia production could be altered by inhibition of the urease enzyme and variations in temperature.

## MATERIALS AND METHODS

### Medical record review

We performed a retrospective cohort study using data from the Emory data warehouse between 2013 and 2022, which was approved by Emory IRB (STUDY00005388). Patients with positive blood or cerebrospinal fluid (CSF) cryptococcal cultures and concomitant plasma ammonia levels within 7 d of the culture were included. Disseminated cryptococcal infection (DCI) was defined as the occurrence of a positive blood culture or a positive culture from at least two different sites (4). Patients with ammonia levels beyond 7 d before or after a positive culture were excluded due to difficulty in attributing plasma ammonia levels to the cryptococcal infection. The 7-d cutoff was arbitrarily agreed by the clinicians in the study. To determine if patients had acute and chronic liver disease when plasma ammonia was measured, a review of individual patients was performed by two study clinicians focusing on chart review of clinician notes, laboratory findings, and abdominal imaging. By employing modifications to an existing criterion (5, 6), patients were designated as having liver dysfunction if a current diagnosis of cirrhosis or acute liver failure was determined by clinical team on chart review and/or two of the three following parameters were met: bilirubin > 2.5 mg/dL, alanine aminotransferase (ALT) ≥ twice the upper limit of normal (≥104 U/L), and international normalized ratio (INR) ≥ 1.5. Patients with prior history of liver failure who had undergone liver transplant and had normal synthetic liver function were designated as having no liver dysfunction.

Two groups were compared based on the presence of hyperammonemia defined by more than 53 µmol/L, which is the upper limit of normal based on our laboratory assay. Renal failure was defined as KDIGO AKI stage 2 (7) and/or CKD stage 4 (8). We used Wilcoxon rank-sum, Fisher's exact, Kruskal-Wallis, Dunn's pairwise comparison with Bonferroni adjustment, and Spearman's correlation tests to determine risk factors for hyperammonemia, which included liver failure, initial cryptococcal titers, DCI, and admission creatinine. DCI was defined as positive blood culture irrespective of CSF culture positivity.

### Laboratory studies

#### C. neoformans culture

The Emory Investigational Clinical Microbiological Core (ICMC) biorepository, which is an IRB approved biorepository (STUDY00093057), was queried for cryptococcal isolates from

patients included in our study. Of the 29 patients, cryptococcal cells from three patients were available and recoverable for analysis and therefore representing a convenience sample. Four sequential isolates were available for patient A, while a single isolate was available for patients B and C (Table S1). Wild-type (WT) *C. neoformans* serotype A strain, KN99α, obtained from the Fungal Genetics Stock Center (9), and cryptococcal cells isolated from patients with hyperammonemia were recovered from frozen 50% glycerol stocks by growth in yeast extract, peptone, glucose (YPD) broth for 2 d at 30°C before use in experiments.

## Urease assay

Phosphate-buffered saline (PBS)-washed cells were sub-cultured in triplicate into urea broth at a density of $2 \times 10^7$ cells/mL and were incubated at either 30°C or 37°C with orbital shaking at 120 rpm. Samples of each culture were centrifuged at $10,000 \times g$ for 5 min, and absorbance at 560 nm ($A_{560}$) of cell-free supernatants as well as media-only controls were read using a Spectramax iD5 plate reader (Molecular Devices). $A_{560}$ readings were blank subtracted and corrected for slight differences in culture densities by quantifying colony forming units (CFU) of $5 \times 10^{-6}$ dilutions plated on Sabouraud (SAB) dextrose agar.

## Ammonia production assay

Urea broth in well B2 of a 12-well plate was inoculated with WT or A1 cryptococcal cells at a density of $5 \times 10^7$ cells/mL in the absence or presence of 5 mM acetohydroxamic acid (AHA). Following addition of 1 mL cell-free pH-indicative buffer (10 mM $KH_2PO_4$, 0.03 mM phenol red) to wells A2, B1, B3, and C2, plates were sealed with parafilm and were incubated at 30°C or 37°C with orbital shaking at 120 rpm. The same buffer was titrated with $NH_4OH$, $A_{560}$ readings for each concentration were plotted against log [$NH_3$], and the resulting curve was fit to a Boltzmann sigmoidal equation, $y =$ bottom + [(top-bottom)/1 + exp($V_{50} - x$/slope)], using GraphPad Prism software. $A_{560}$ readings of each buffer-containing well measured over the course of several hours were blank subtracted and CFU corrected (as described above) and were used to extrapolate the concentration of ammonia ($NH_3$) in the buffer using the standard curve. Plates were photographed after incubation for 6 h using a 12-megapixel camera.

## Growth curves

WT *C. neoformans* and patient isolates were sub-cultured from YPD into urea-supplemented minimal media (MM) consisting of 29.4 mM $K_2HPO_4$, 10 mM $MgSO_4$, 13 mM glycine, 15 mM D-glucose, 3 µM thiamine, and 10 mM urea at pH 5.5 at a density of $1 \times 10^6$ cells/mL in wells of a 48-well plate. The plate was incubated at 30°C with orbital shaking in a Spectramax iD5 plate reader, and $A_{600}$ was read at 15-min intervals. Growth curves were plotted as $A_{600}$ against time, and rates were derived from a linear regression fit of the linear portion of each plot using GraphPad Prism software. Following centrifugation for 5 min at $2,500 \times g$, the pH of each culture supernatant was measured using an Accumet AB150 pH meter.

## Microscopy

Unstained or India ink-stained live cells from stationary phase cultures grown in YPD or MM, respectively, were imaged using phase contrast illumination with an oil immersion 100× objective on an Olympus AX70 microscope. Images were captured using QCapture Pro software with a Retiga 1300 digital charge-coupled device (CCD) camera. Pixel dimensions for cell bodies and capsules were measured from images using Adobe Photoshop and then converted to micrometers using the conversion factor of 0.0645 µm/pixel.

## Melanin production

MM agar with or without the addition of 1 mM dopamine was dispensed into wells of a 48-well plate and was allowed to solidify at room temperature (RT) before being spotted with $1 \times 10^6$ PBS-washed cells. Plates were photographed after incubation at 30°C for 24, 48, and 72 h. Color images were converted to grayscale using Adobe Photoshop, and relative pigment intensities were quantified using ImageStudio Lite software. The signal measured for WT cells grown on MM agar without dopamine was used for background subtraction.

## M13 molecular typing

Genomic DNA was isolated from WT and patient isolates using the hexadecyltrimethy-lammonium bromide (CTAB) extraction method (10). Briefly, cells were harvested by centrifugation after growth in YPD, then frozen on dry ice and lyophilized overnight. After being vortexed with 0.5-mm glass beads, powdered cell pellets were incubated for 30 min at 65°C in lysis buffer consisting of 100 mM Tris, pH 7.4, 700 mM NaCl, 10 mM EDTA, 1% CTAB, and 1% beta-mercaptoethanol. Following extractions with chloroform and isopropanol, DNA pellets were washed with 70% ethanol, resuspended with sterile water, and then treated with 20 µg RNase A for 30 min at 37°C.

Molecular typing of genomic DNA preparations was performed using a modification of the Meyer et al. protocol (11). PCR mixtures consisted of 1 µg genomic DNA, 100 ng M13 primer, 0.2 mM of each nucleotide (dATP, dCTP, dGTP, and dTTP), 1.5 mM $MgCl_2$, and 2 units of Phusion DNA polymerase (M0535S) in the provided GC buffer. An initial denaturation for 2 min at 94°C was followed by 35 cycles of denaturation for 20 s at 94°C, annealing for 1 min at 50°C, and an extension for 20 s at 72°C, and a final extension for 6 min at 72°C. PCR products were resolved by electrophoresis in a 1.4% agarose gel in 1× Tris-borate-EDTA buffer supplemented with 0.5 mg/mL ethidium bromide and were imaged under UV light using a Bio-Rad ChemiDoc XRS System.

## Sanger DNA sequencing of urease ORF

The urease open reading frame (ORF) was PCR amplified from cDNA synthesized from total RNA extracted from WT and patient isolate cultures using a Bio-Rad T100 Thermal Cycler and forward and reverse primers 5′-GGACACGAGCACGGATACAC and 5′-GTTCATGA ATAATGGAGAAATGCAC, respectively. Sanger DNA sequencing reactions were conducted at the Genetic Resources Core Facility, Johns Hopkins Department of Genetic Medicine, Baltimore, MD.

## Total RNA isolation and cDNA synthesis

WT *C. neoformans* and patient isolates recovered from frozen 50% glycerol stocks were grown in three independent cultures in YPD for 2 d at 30°C, and then $1 \times 10^8$ PBS-washed cells from each culture were harvested or sub-cultured into MM for 24 h at 30°C before being harvested. Cell pellets that had been frozen on dry ice and stored at −80°C were thawed in 1 mL Trizol and were lysed using Lysing Matrix C beads (MP Biomedicals) in a Fisherbrand Bead Mill 24 homogenizer with 4 pulses of 30 s at speed setting 6. Total RNA was extracted from cell lysates using a PureLink RNA Mini kit (Thermo Fisher Scientific) according to the manufacturer's protocol and including on-column DNAse treatment. RNA yields were quantified using a Qubit Broad Range Assay kit in a Qubit 4 Fluorometer (Thermo Fisher Scientific), and then 1 µg total RNA from each sample was used as a template for cDNA synthesis using a SuperScript IV First Strand Synthesis System (Invitrogen) with the provided oligo (dT)20 primer.

## Quantitative real-time PCR

Real-time PCR in a Step One Real-Time PCR System (Applied Biosystems) was used to quantify the urease gene product (*URE1*) relative to the endogenous housekeeping gene,

actin (*ACT1*), expressed by WT and patient isolates grown in rich (YPD) or low glucose (MM) medium as described above. PCR mixtures consisting of SYBR Green Master Mix (Applied Biosystems), 2 µL template cDNA, and 0.2 µM each primer in a 20-µL volume were denatured for 10 min at 95°C, followed by 40 cycles of denaturation for 15 s at 95°C and annealing/extension at 60°C for 1 min. A melting curve (60°C to 95°C with 15 s reads) was included to confirm the amplification of a single PCR product in each reaction. Primer sequences (5′ to 3′) were as follows: *URE1* (forward) TCGTATCGGTGAAG TCGTCACT, *URE1* (reverse) GGACCACGGAATTGCTTCAT, *ACT1* (forward) CCACACTGTCCCC ATTTACGA, and *ACT1* (reverse) CAGCAAGATCGATACGGAGGAT. Relative gene expression was calculated using the $2^{\Delta\Delta CT}$ method (12).

## Urease activity in cell lysates

WT *C. neoformans* and patient isolate A1 cells were sub-cultured in triplicate from YPD into MM at a cell density of $1 \times 10^7$ cells/mL and were incubated at 30°C for 3 d. Cells were harvested by centrifugation at $3,000 \times g$ for 15 min, washed with PBS, then resuspended in lysis buffer consisting of 10 mM sodium phosphate, pH 7.0, and 1× Complete Protease Inhibitor Cocktail (Roche). Cells were chilled on ice then lysed by two passes through a French Press High Pressure Cell Disrupter (Glen Mills) set at 24,000 psi, followed by centrifugation at $3,000 \times g$ for 15 min at 4°C to remove cell debris. Cleared lysates were diluted 20-fold in lysis buffer and were assayed for total protein content using a Micro BCA Protein Assay kit (Pierce) following the manufacturer's instructions. Urease activity in lysates was quantified by the Berthelot method using a Urease Activity Assay kit (Sigma, MAK120) according to the provided protocol. Ammonia measurements were corrected for the total protein content in each lysate.

## RESULTS

### Medical record review

Our medical record review revealed 29 patients with cryptococcal infection, comprising 28 cases of *C. neoformans* and 1 case of *C. gattii*. Among these patients, 37.9% had hyperammonemia, 59% had DCI, and 41% had isolated central nervous system (CNS) cryptococcal infection (CNS-CI). Additionally, 38% of patients had renal failure and 28% had liver disease (Table 1). Renal failure was associated with 4.4 times (95% confidence interval [CI] 1.5, 13.0) higher risk of hyperammonemia. This risk was found to be higher among patients with DCI (RR 6.2, 95% CI 1.0, 40.2) as compared to those without DCI (RR 2.5, 95% CI, 0.40, 16.0) (Table 2). There was a moderate correlation between ammonia levels and admission creatinine (Spearman rho 0.40; $P = 0.03$) (Fig. 1). Liver disease and cryptococcal titers were not associated with the presence of hyperammonemia. The small sample size precluded assessment of co-linearity between liver failure and renal failure. Overlap of renal and liver failure accounted for higher median plasma ammonia level of 93 µmol/L (25–242, $n = 5$) compared to liver disease alone 37 µmol/L (16–145, $n = 3$) or renal failure alone 66 µmol/L (48–692, $n = 6$) ($P = 0.0204$) (Table 3). Post-hoc analysis using Dunn's test shows significant pairwise association only between none and renal failure only (Table 4). The occurrence of severe hyperammonemia >200 µmol/L was uncommon, occurring in only two patients, one with cirrhosis and renal failure with a peak level of 242 µmol/L, and the other was an outlier with renal failure in a recent liver transplant recipient with a plasma ammonia level of 692 µmol/L and normal liver function. Other causes of hyperammonemia were excluded in this patient, raising the possibility of increased urease activity unique to the strain.

**TABLE 1** Sample characteristics of patients enrolled in cryptammonia study with stratification by the presence of hyperammonemia, Emory data warehouse, 2013–2022 (N = 29)

| Characteristics (continuous) | Overall (n = 29) | Hyperammonemia absent (n = 18) | Hyperammonemia present (n = 11) | P value |
|---|---|---|---|---|
| | | Median (IQR)[b] | | Wilcoxon rank-sum test |
| Age | 63 (54–68) | 63 (56–66) | 63 (48–73) | 0.530 |
| Admission BUN[a] | 26 (19–43) | 20 (16–34) | 38 (21–57) | 0.548 |
| Preadmission creatinine (n = 21) | 1.3 (0.7–1.7) | 1.3 (0.6–1.7) | 1.3 (0.7–2.3) | 0.423 |
| Admission creatinine | 1.4 (0.6–1.8) | 1.0 (0.6–1.8) | 1.6 (1.0–2.9) | 0.331 |
| Peak creatinine | 1.5 (0.8–2.5) | 0.9 (0.6–1.8) | 2.3 (1.7–3.0) | 0.167 |
| Admission BUN–creatinine ratio | 20 (15–26) | 21 (17–24) | 18 (14–27) | 0.313 |
| Length of stay (days) | 18 (9–27) | 15 (9–20) | 18 (8–75) | 0.394 |
| Blood titer | 2,560 (320–2,560) | 2,304 (320–2,560) | 2,560 (318–2,560) | 0.429 |
| CSF titer | 1,280 (160–2,560) | 1,820 (160–2,560) | 1,280 (1,280–2,560) | 0.493 |
| **Characteristics (categorical)** | | N (%) | | Fisher's exact test |
| Male gender | 20 (69) | 12 (67) | 8 (73) | 1.000 |
| Liver disease preceding NH$_3$ increase | 8 (28) | 3 (17) | 5 (45) | 0.197 |
| Acute kidney injury | 11 (38) | 3 (17) | 8 (78) | 0.005 |
| Immunosuppression (%) | 16 (55) | 11 (61) | 5 (45) | 0.466 |
| Died (%) | 20 (69) | 11 (61) | 9 (82) | 0.412 |
| Blood/CSF culture positivity: | | | | |
| Isolated blood | 9 (31) | 3 (17) | 6 (55) | |
| Isolated CSF | 2 (41) | 9 (50) | 3 (27) | 0.120 |
| Both | 8 (28) | 6 (33) | 2 (18) | |

[a]BUN, blood urea nitrogen.
[b]IQR, interquartile range.

## Laboratory studies of cryptococcal isolates

### Elevated extracellular urease activity for *C. neoformans* isolated from a patient with hyperammonemia

We sought to investigate whether hyperammonemia in patients with cryptococcal infections could be associated with increased urease activity of the fungal cells. Patient isolates and the laboratory *C. neoformans* strain, KN99α, were grown in pH-indicative urea broth (13), and relative extracellular urease activity was quantified by measuring the absorbance at 560 nm ($A_{560}$) of culture supernatants. At both the preferred cryptococcal growth temperature of 30°C and the human body temperature of 37°C, isolate A1 from the patient with the most rapidly increasing plasma ammonia level, peaking at 692 µmol/L (Table S1), had 4- to 5-fold higher urease activity than the WT reference strain (Fig. 2a). Although four isolates (A1–A4) had been collected from the same patient on different days, isolate A1 showed significantly higher urease activity than the other three isolates from the same patient or from the other two patients (Fig. 2a).

Molecular type analysis revealed banding patterns that differed from the WT reference strain for isolates from patients A and C (Fig. S1a). Furthermore, the M13 fingerprint of isolate A1 was distinct from that of the other three patient A isolates (Fig. S1a). In light of this evidence for genomic rearrangements, we sequenced the urease open reading frame in our WT laboratory strain and patient isolates to identify sequence polymorphisms (Fig. S1b). Other than a short region of sequence ambiguity near the 5′

**TABLE 2** Univariate risk estimated of renal failure alone and with/without DCI, cryptammonia study, Emory data warehouse, 2013–2022 (N = 29)

| Renal failure with/without DCI | Sample size | Risk ratio (95% CI) |
|---|---|---|
| Renal failure irrespective of DCI | 29 | 4.4 (1.5, 13.0) |
| Renal failure with DCI | 17 | 6.2 (1.0, 40.2) |
| Renal failure without DCI | 12 | 2.5 (0.4, 16.0) |

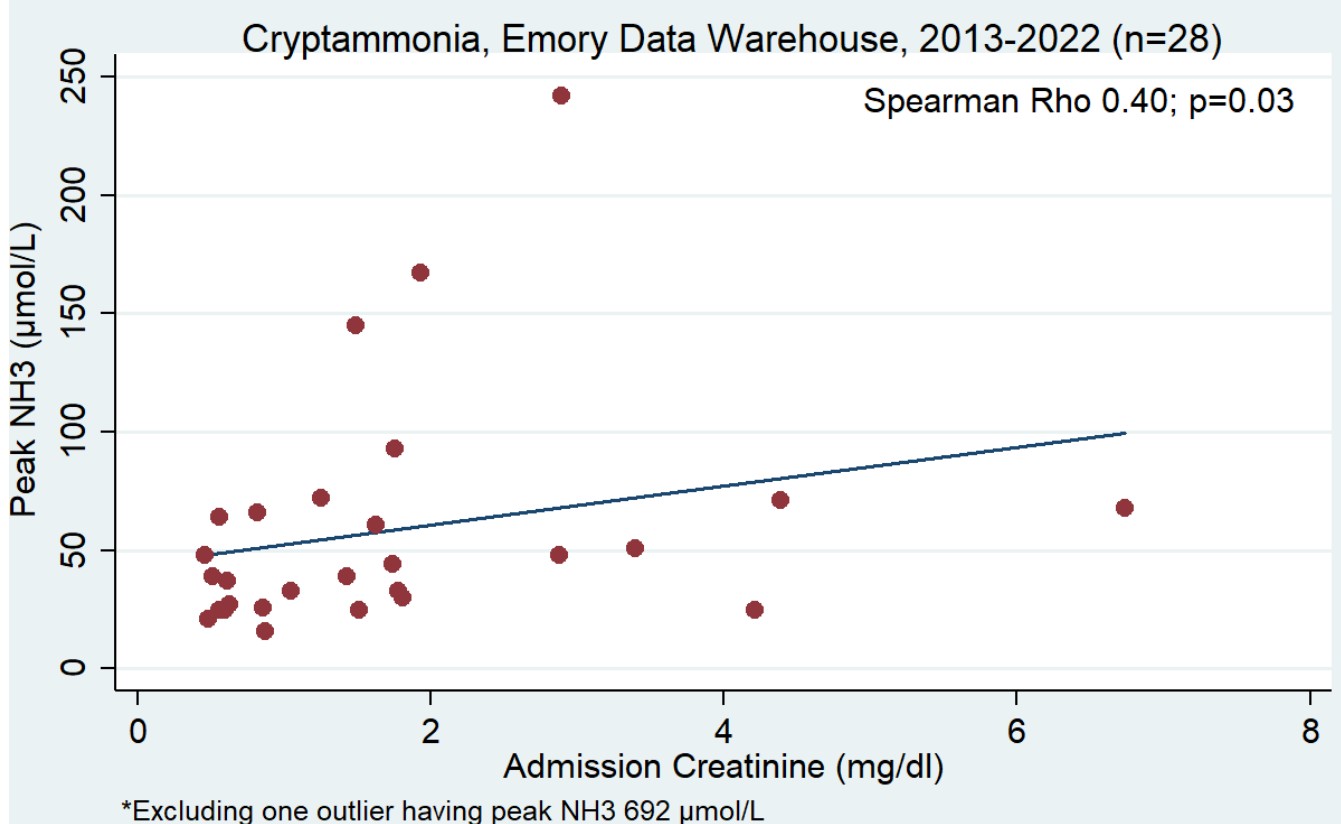

**FIG 1**  Scatterplot with a linear-fitted line showing the relationship between admission creatinine and peak plasma ammonia level (NH$_3$). There was a moderate correlation between ammonia levels and admission creatinine (Spearman rho 0.40; $P = 0.03$).

end, only a few sequence changes from the H99 reference sequence were noted in the remainder of the open reading frame. Patient isolates A1, A2, A3, A4, and C shared C to T transitions at 271 and 1,287 resulting in silent mutations of L91 and S429, respectively, and a G to A transition at 750 resulting in a silent mutation of R250. Another C to T transition at 2,643 producing a silent mutation of Y881 was detected in patient isolates A1, A2, A3, and B and the WT KN99α strain. An amino acid change of V772M resulting from a G to A transition at 2,314 was present in the WT KN99α strain and patient isolates A1, A2, A3, and B.

To derive a quantitative measure of the amount of ammonia produced by isolate A1 compared to WT, *C. neoformans* cells were grown in urea broth in one well of a 12-well plate, and $A_{560}$ of cell-free pH-sensitive buffer in surrounding wells (Fig. 2b) was monitored over time. The amount of ammonia that had diffused into each well was extrapolated from a standard curve of ammonia versus $A_{560}$ (Fig. 2c). Average ammonia measurements for the A1 isolate of 37 and 78 nM per $1 \times 10^6$ cells per hour at 30°C and 37°C, respectively, were approximately twofold greater than those measured for

**TABLE 3**  Median peak NH$_3$ by the presence of renal failure or liver disease, cryptammonia study, Emory data warehouse, 2013–2022 ($N = 29$)

| Renal failure or liver disease | Sample size | Median peak NH$_3$ (range) | P value |
|---|---|---|---|
| No renal failure and liver disease | 15 | 33 (21–72) | 0.0204 |
| Liver disease, no renal failure | 3 | 37 (16–145) | |
| Renal failure, no liver disease | 6 | 66 (48–692) | |
| Both renal failure and liver disease | 5 | 93 (25–242) | |

TABLE 4 Dunn's pairwise comparison with Bonferroni adjustment (*P* value) of peak NH₃ and presence of renal failure or liver disease, cryptammonia study, Emory data warehouse, 2013–2022 (*N* = 29)

| Dunn's pairwise comparison with Bonferroni adjustment (*P* value) | No renal failure and liver disease | Liver disease, no renal failure | Renal failure, no liver disease |
|---|---|---|---|
| Liver disease, no renal failure | 1.0000 | | |
| Renal failure, no liver disease | 0.0292 | 0.4844 | |
| Both renal failure and liver disease | 0.0569 | 0.5760 | 1.0000 |

the equivalent number of WT cells (Fig. 2d). Ammonia production by isolate A1 was as effectively inhibited by the FDA-approved urease inhibitor, AHA, as the WT strain, to about 5% of untreated levels at both temperatures (Fig. 2e).

### Hyperammonemia patient A isolates exhibit slow growth and enlarged cell size phenotypes

Upon microscopic examination of cells grown for urease activity assays, cells from the strain derived from hyperammonemia patient A appeared enlarged compared to WT and the other patient isolates. To assess this phenotype quantitatively, cell radii measurements were made from images of more than 40 cells from each stationary phase culture, and this analysis revealed that hyperammonemia isolate A1 cells were significantly larger ($P < 0.0001$) than WT or isolates from patients B and C (Fig. 3a). Cell measurements were also significantly larger for two of the three sequential isolates, A3 and A4, from patient A (Fig. S2a). Although not larger on average than WT cells, some of the cells in cultures of isolate A2 displayed a budding defect in the form of pseudo-hyphal growth (Fig. S3).

Comparison of growth rates in MM revealed the most significantly reduced growth rate for patient isolates A1 and A4, and a somewhat slower growth rate for patient isolates A2, A3, and C compared to WT (Fig. 3b; Fig. S2b). As *C. neoformans* growth is disfavored at high pH (14), increased alkalinity resulting from increased ammonia production by patient isolates may explain their decreased growth rate. Notably, growth was decreased to comparably low rates for WT and the three patient isolates when growth media was supplemented with 5 mM AHA (Fig. 3b) and was significantly slower in the presence, compared to the absence, of AHA for all strains (Fig. S2c). The pH of culture supernatants measured at the completion of the growth experiment was also significantly lower for cultures grown in the presence, compared to the absence, of AHA (Fig. S2c), which is consistent with decreased ammonia production due to urease inhibition. The growth inhibitory effect of AHA is consistent with previous reports of decreased *in vitro* growth rates for urease-deficient *C. neoformans* (15, 16) and underscores the potential therapeutic benefit of urease inhibition in the clinical setting.

### Two virulence factors, capsule and melanin production, are diminished in hyperammonemia patient isolates

Capsule enlargement is typically induced by growth in nutrient-limited media, but after 3 d in MM, isolates from patients A and C exhibited significantly increased cell body sizes and decreased capsule sizes compared to WT (Fig. 4a; Fig. S4a). Interestingly, the opposite trend was observed for patient isolate B, which had smaller cells and larger capsules than WT (Fig. 4a).

Melanin production by patient isolates A, B, and C occurred at a reduced rate compared to WT (Fig. 4b; Fig. S4b). The defect was less severe for isolates A2, A3, B, and C, as WT levels of pigment were produced after 72 h (Fig. 4b; Fig. S4b), while isolates A1 and A4 required a week to become fully pigmented (data not shown).

### Urease is not overproduced by hyperammonemia patient isolates

Urease expression was compared for WT and hyperammonemia patient isolates by quantitative real-time PCR, normalized to the housekeeping actin gene, *ACT1*. Expression

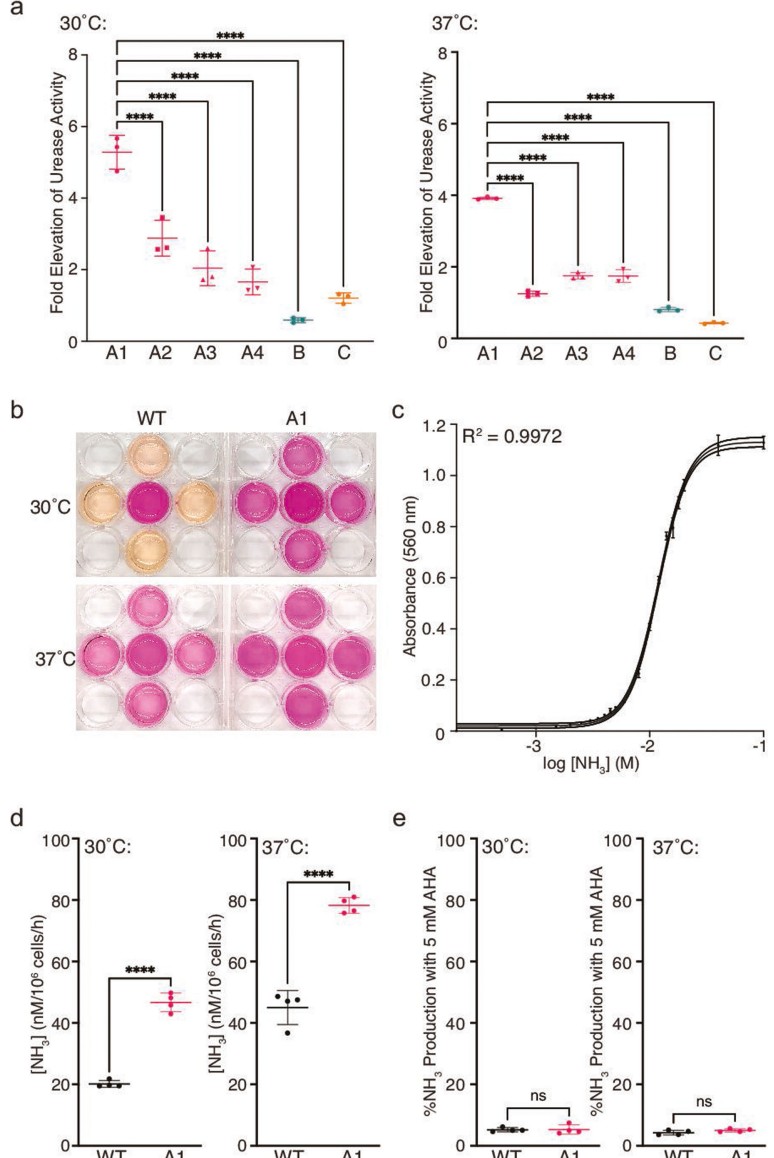

**FIG 2** Hyperammonemia patient isolate A1 displays significantly elevated extracellular urease activity and ammonia production that can be effectively inhibited by AHA. (a) Quantified extracellular urease activity relative to WT for hyperammonemia isolates revealed significantly higher activity for A1 compared to the other five isolates at both 30°C (left graph) and 37°C (right graph). **$P = 0.0014$, ****$P < 0.0001$; ordinary one-way analysis of variance (ANOVA) with Tukey's multiple comparisons test. (b) Color photographs of WT and A1 cells grown in urea broth (center well) with cell-free pH-responsive buffer in four surrounding wells following incubation at the indicated temperatures for 6 h. (c) Standard curve of ammonia ($NH_3$) versus $A_{560}$ of pH-sensitive buffer fit using nonlinear regression analysis with a Boltzmann sigmoidal equation. (d) Quantification of $NH_3$ concentration in cell-free buffer wells by extrapolation from the standard curve. Measurements are calibrated for $10^6$ WT or A1 cells per hour at the indicated temperatures. ****$P < 0.0001$; unpaired, two-sided parametric $t$-test. (e) Percent $NH_3$ produced by WT and A1 cells in the presence of 5 mM urease inhibitor, AHA, compared to untreated controls at 30°C and 37°C. ns = not significant; unpaired, two-sided parametric $t$-test.

levels were comparable for cultures grown in rich media (Fig. 5a). Urease expression in minimal media was significantly decreased compared to WT for patient isolates A1 and A4 (Fig. 5b), and notably, these were the same isolates that displayed the most severe

defect in melanin production (Fig. S4b). Comparison of the relative urease expression levels in minimal compared to rich media revealed an average twofold increase in urease expression for WT and patient isolates upon glucose deprivation (Fig. 5c). These data are consistent with a previous report of higher urease activity measured in cell lysates of *C. neoformans* cultures grown in nutrient-limited compared to rich media (15). We compared urease activity in cell lysates prepared from WT and patient isolate A1 minimal media cultures and found reduced intracellular urease activity for the hyperammonemia patient isolate (Fig. 5d), which is consistent with lower urease expression measured in minimal media (Fig. 5b). Hence, increased urease-mediated ammonia production by the hyperammonemia patient isolate cannot be attributed to increased urease production within the cell, but more likely results from increased secretion of urease from the cell.

## DISCUSSION

Severe hyperammonemia can lead to devastating neurological consequences including encephalopathy, cerebral edema, seizure, coma, and brain herniation (17). Urease-producing pathogens like *Ureaplasma urealyticum* have been identified as a cause of non-hepatic hyperammonemia in immunocompromised hosts with an associated high mortality rate due to neurological injury (18). Urease is an enzyme that catalyzes the

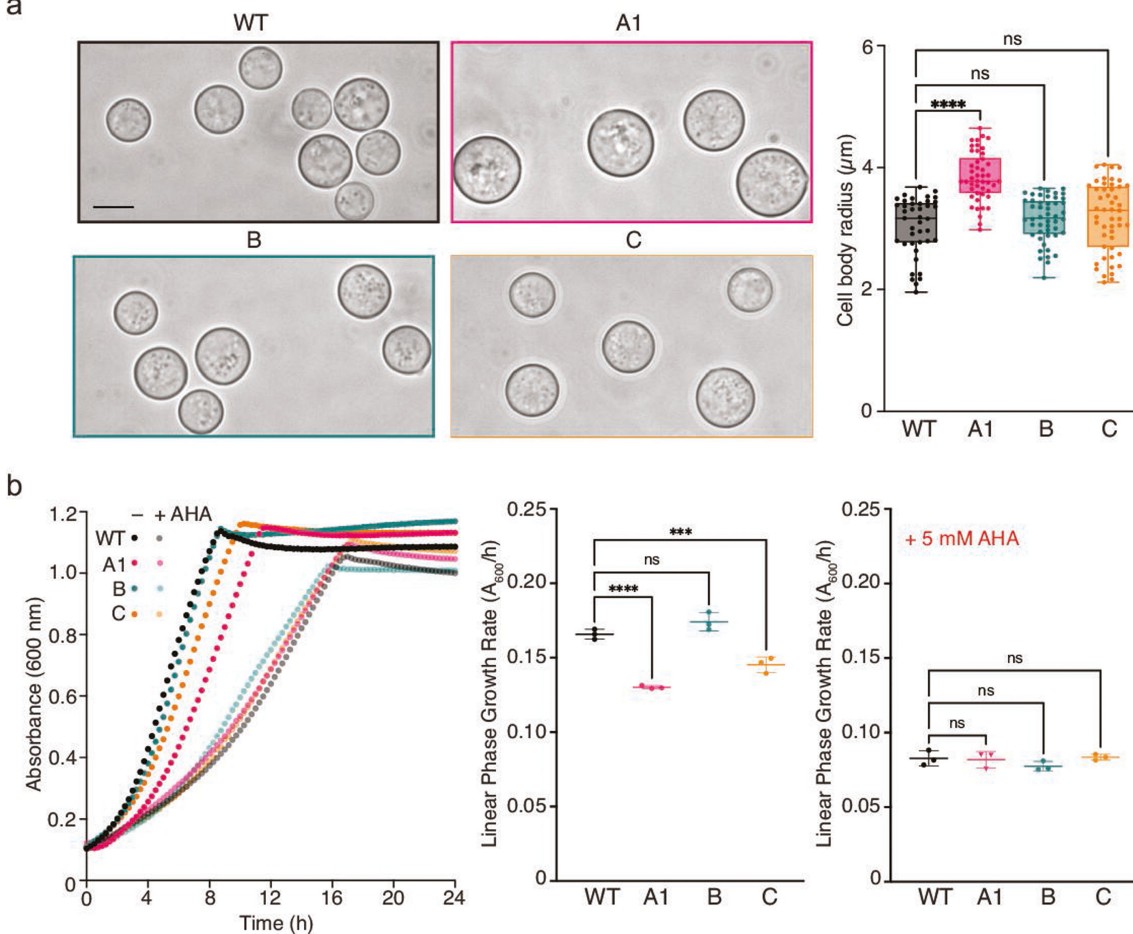

**FIG 3** Increased cell size and reduced growth rate of isolate A1 compared to WT. (a) Representative microscopy images of *C. neoformans* cells grown to stationary phase in rich media (left panels), scale bar is 5 µm. Quantification of cell body radii (right panel) shows a significantly larger average cell body size for isolate A1 compared to WT. ns = not significant, ****$P$ < 0.0001; ordinary one-way analysis of variance (ANOVA) with Tukey's multiple comparisons test. (b) Growth of WT and hyperammonemia isolates at 30°C in 10 mM urea-supplemented minimal media ± 5 mM AHA plotted as $A_{600}$ against time (left panel) and linear phase growth rates (right panels) extrapolated from the linear portion of each growth curve. ns = not significant, **$P$ = 0.0020, ****$P$ < 0.0001; ordinary one-way ANOVA with Tukey's multiple comparisons test.

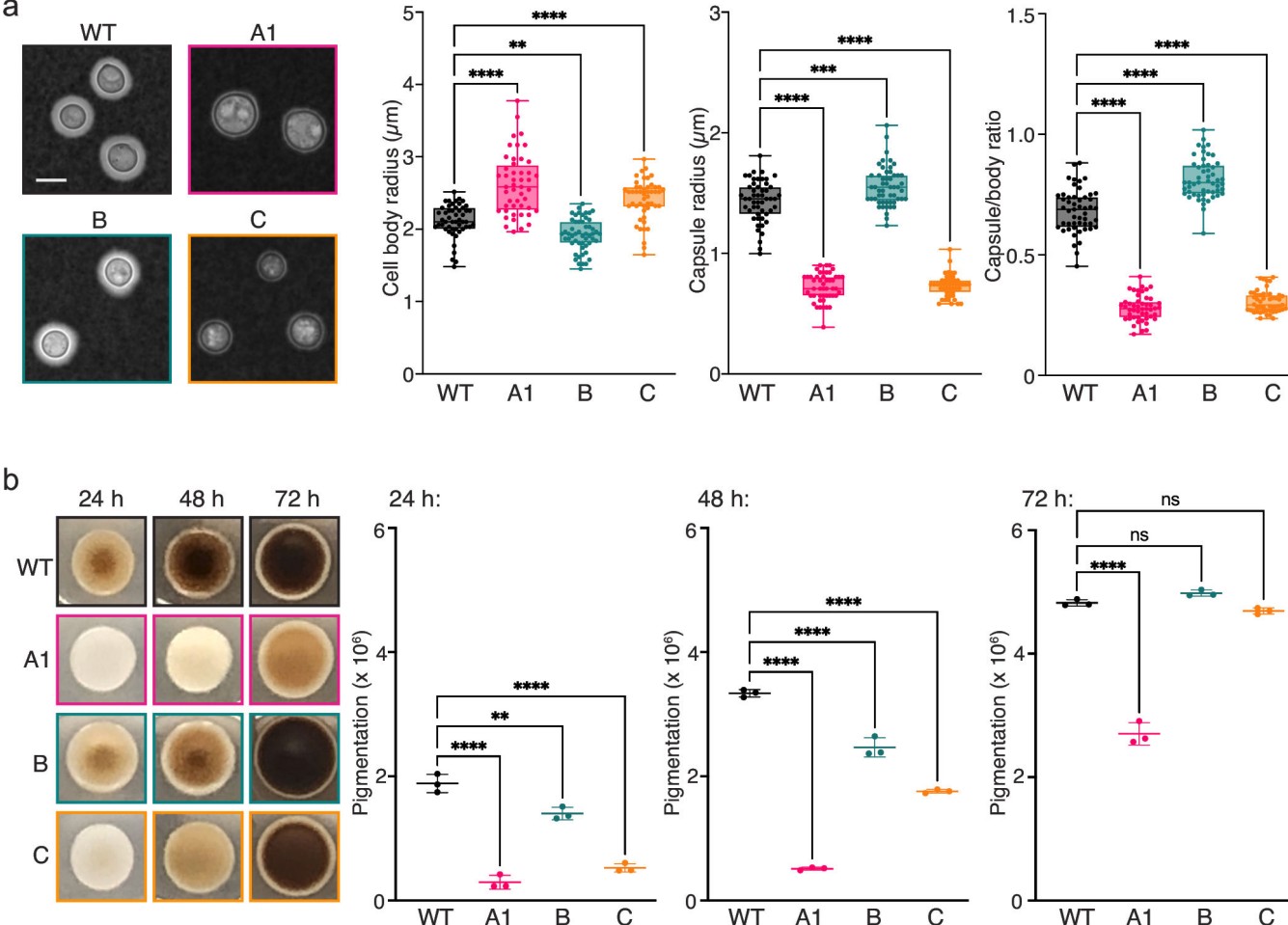

**FIG 4** Impaired capsule and melanin synthesis of hyperammonemia patient isolates. (a) Representative microscopy images of cells stained with India ink after growth in low glucose for 3 d (left panels), scale bar is 5 µm. Quantification of cell body and capsule sizes (graphs, right panels) reveals a significant increase in cell size for isolates A1 and C, and a concomitant decrease in capsule size compared to WT, whereas isolate B shows the opposite trend. $**P = 0.0049$, $***P = 0.0001$, $****P < 0.0001$; ordinary one-way analysis of variance (ANOVA) with Tukey's multiple comparisons test. (b) Color photographs (left panels) of *C. neoformans* cells grown on dopamine-supplemented agar for the indicated amount of time (each panel is 4 cm²). Quantified pigmentation (right panels) showing delayed pigment production compared to WT for the hyperammonemia isolates with the most severe delay observed for isolate A1. ns = not significant, $**P = 0.0029$, $****P < 0.0001$; ordinary one-way ANOVA with Tukey's multiple comparisons test.

hydrolysis of urea to ammonia and carbon dioxide. Recognition and treatment of *U. urealyticum* with appropriate antimicrobial therapy and control of plasma ammonia levels have resulted in a paradigm shift with improved survival (19). Other well-known organisms like *Proteus*, *Klebsiella*, *Helicobacter pylori*, and *C. neoformans* and *gattii* are also urease producers (20). Cryptococcal infections remain a significant cause of morbidity and mortality globally with significant neurological disease burden. The occurrence and influence of hyperammonemia in cryptococcal infections remain largely unexplored. Our study suggests that hyperammonemia does occur in the setting of cryptococcosis and may be driven by three factors: (i) renal failure, (ii) disseminated infection, and (iii) increased urease secretion unique to the cryptococcal strain.

Renal failure was associated with 3.8 times (95% CI 1.2, 11.8) higher risk of hyperammonemia. This risk was found to be higher among patients with DCI (RR 5.3, 95% CI 0.8, 35.3) than those without DCI (RR 2.5, 95% CI 0.40, 16.1). The association between renal failure and hyperammonemia in cryptococcal infections appears robust and even surpasses the association with liver failure/dysfunction. This finding, however, is reliant on an assumption that patients with renal failure without cryptococcal infections have

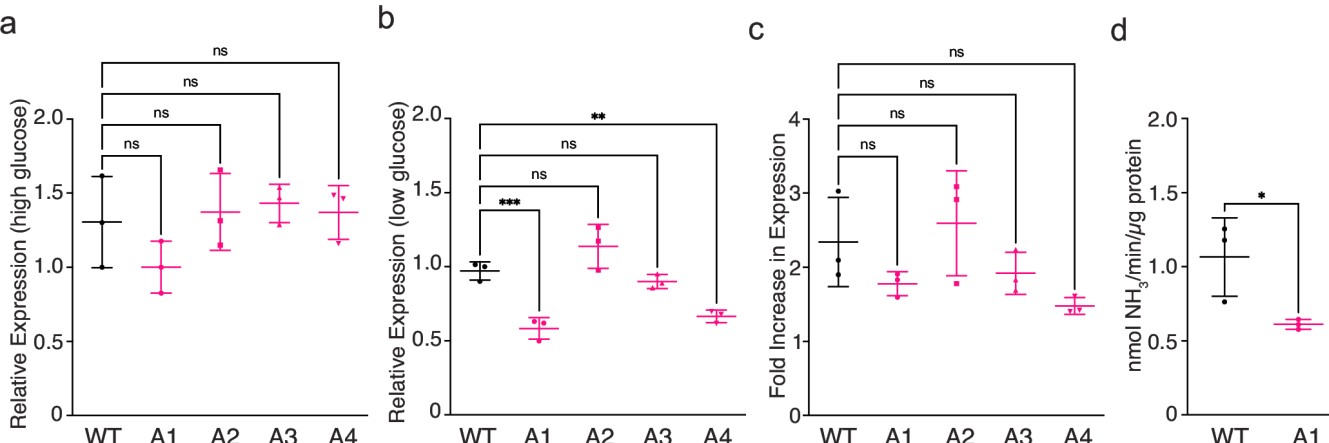

**FIG 5** Comparison of urease expression levels in WT and patient A isolates. Expression of urease relative to the actin housekeeping gene in three biological replicates of WT and four sequential isolates from patient A grown in rich media (a) or minimal media (b). In each case, data are normalized to one of the three WT samples with mean and standard deviation indicated by error bars. (c) Relative urease expression in minimal media compared to rich media expressed as the fold increase. (d) Urease activity measured in cell lysates from WT and patient A1 isolate cultures grown in minimal media. Measurements are normalized to total protein content. ns = not significant, $**P = 0.0039$, $***P = 0.0007$; ordinary one-way analysis of variance (ANOVA) with Tukey's multiple comparisons test.

normal plasma ammonia levels. In a study looking at 23 hemodialysis patients, pre-dialysis plasma ammonia level was 21.7 ± 1.2 µmol/L (21), suggesting that plasma ammonia levels are usually normal despite renal failure (CKD stage 5). In cryptococcal infections, the association between renal failure and hyperammonemia may be attributable to the inability of the kidneys to eliminate ammonia either directly or indirectly as urea or glutamine. Alternatively, uremia in renal failure may serve as a substrate for the cryptococcal urease enzymatic activity, leading to hyperammonemia. This association between uremia and ammonia production has been demonstrated experimentally as urease-producing *U. urealyticum* and *Ureaplasma parvum* grown in uremic conditions resulted in increased ammonia production attributable to increased urease activity (22). However, establishing a direct correlation between ammonia levels and urea levels in a clinical setting can be challenging. This is primarily due to the complex interplay of various host processes that regulate serum urea levels, as well as the fact that urea is consumed in this reaction.

The pathogenesis of *C. neoformans* is influenced by a number of virulence factors, which include the polysaccharide capsule, the production of melanin, thermotolerance, and the production of extracellular enzymes such as laccase, urease, and phospholipases. Urease enhances the ability of *Cryptococcus* to cross the blood–brain barrier. Although *C. neoformans'* growth is favored by a low pH environment, increase in pH from ammonia production slows down replication and enhances melanization, which augments its ability to survive within phagosomes and disseminate into the brain and cerebrospinal fluid using a Trojan horse mechanism (23). Clinical manifestation commonly associated with cryptococcal infections include encephalopathy, cerebral edema, coma, and brain herniation. The association between hyperammonemia and cryptococcal infections raises important considerations. Elevated ammonia levels could accelerate the clinical deterioration of patients, exacerbating neurological symptoms and potentially contributing to the poor outcomes observed in severe cases (1–3).

Our laboratory studies showed that the initial strain obtained from patient A (isolate A1) when plasma ammonia level was increasing rapidly (Table S1) had a significant increase in both extracellular urease activity and ammonia production compared to wild-type cells. This may explain why the ammonia production in this patient was an outlier. Interestingly, subsequent isolates from this patient obtained on days 3, 5, and 12 on antifungal therapy revealed a reduction in urease activity. *C. neoformans* is well known for undergoing microevolution in the human host as minor karyotype

differences have been detected in sequential isolates from the same patient (24), and these polymorphisms in serial patient isolates correlate with altered virulence traits (25). Indeed, the phenotypic differences observed for sequential isolates from patient A may have resulted from genomic rearrangements during infection because molecular typing revealed a unique M13 fingerprint for isolate A1 compared to the other three isolates from the same patient (Fig. S1a). Only a few silent mutations were found in the urease open reading frame so increased ammonia production by the patient isolates is not likely attributable to a change in the urease enzyme itself. Furthermore, it is important to note that the underlying genetic basis for the observed phenotypes is likely multifactorial, and could suggest a mixed infection in patient A. This highlights the complexity and diversity of cryptococcal infections, with approximately 20% of cases involving the presence of more than one strain (26).

To further characterize differences in the strain, we assessed strain morphology, growth rates, and virulence factors. Patient isolate A1 was morphologically larger with significantly larger cell radii and demonstrated a slower growth rate compared to WT or strains B and C. Patient A sequential isolates showed a similarly decreased growth rate as well as enlarged cells for A3 and A4 and a budding defect for A2. These findings imply a potential disruption or delay in the cell division process, leading to a prolonged cell cycle. The significance of a prolonged cell cycle in this context requires further investigation, as it may indicate underlying molecular mechanisms that could have implications in *C. neoformans* virulence and pathogenicity. For instance, faster growth rates are associated with increased incidence of lytic release from phagocytes, while slower intracellular growth rates, which have been correlated with increased urease activity (16), prolong the persistence of cryptococcal cells within the phagosome. The latter, however, is usually accompanied by increased polysaccharide capsule thickness and a melanin pigment layer, which were not apparent in isolate A.

A recent discovery of coordinated expression of individual virulence factors, including urease, polysaccharide capsule, and melanin pigment layer, suggests a sophisticated regulatory system that enables *C. neoformans* to adapt rapidly to changing conditions during host infection (23). Reduction in melanin production and capsule growth was noted in isolates from patients A, B, and C compared to WT. This defect was most pronounced in isolates A1 and A4. Urease gene expression in minimal media was also decreased for these two isolates, and lower intracellular urease activity was measured for isolate A1 compared to WT, which suggests that laccase and urease expression may be regulated by a similar mechanism. Increased ammonia production by hyperammonemia isolates expressing less urease enzyme suggests that urease secretion is a more pertinent factor in ammonia production by *C. neoformans* than urease production. It is plausible that genetic or epigenetic variations affecting the conserved Gα protein/cyclic AMP (cAMP) pathway may have occurred in these isolates as this pathway was found to positively regulate both capsule and melanin production while negatively regulating urease secretion (27, 28). Lower protein kinase A (PKA) activity in the hyperammonemia isolates would explain the increased urease activity and concurrent decrease in capsule and melanin production.

Although plasma ammonia levels were eventually controlled with antifungal therapy in the three reported cases, the severe hyperammonemia was associated with significant morbidity and increased resource utilization. The patients required ventilator support for acute encephalopathy, emergent dialysis, and neuroimaging, and suffered cognitive dysfunction. Inhibition of urease activity represents a logical approach to address hyperammonemia in cryptococcal infections. In the context of *Ureaplasma* infections, the use of a urease inhibitor has been demonstrated to limit the growth of *Ureaplasma* species *in vitro* and *in vivo* in mice and sheep (29, 30), and to resolve *Ureaplasma*-induced hyperammonemia in mice (31). In our study, ammonia production by both isolate A1 with increased extracellular urease activity and WT was effectively inhibited by AHA to about 5% of untreated levels (Fig. 1e). Beyond inhibition of ammonia production, growth media supplemented with 5 mM AHA resulted in a statistically significant reduction in

cryptococcal growth rate across all strains. This is supported by the evidence that genetic deletion of urease results in decreased growth rate (15, 16). This finding suggests that urease inhibitors may hold promise as adjunctive treatments to prevent the occurrence and sequelae of hyperammonemia, and promote the elimination of the organism with antimicrobial therapy by decreasing the growth rate of the organism. As AHA is already an FDA-approved urease inhibitor for chronic urinary tract infections, further preclinical and clinical studies should evaluate its efficacy as an adjunctive therapy for cryptococcal infections.

This study set out to better understand the association of hyperammonemia in cryptococcal infections. Although the retrospective study did not encounter another patient with plasma ammonia levels comparable to patient A, an association with mild to moderate levels of hyperammonemia was apparent especially in the presence of renal failure and disseminated infection. The association of plasma ammonia levels in our study was reliant on chance rather than using a systematic prospective approach, which limits the interpretation of true incidence. Our finding does, however, warrant a systematic prospective study looking at this association. The laboratory analysis of the strains was limited by the availability of only three strains in our repository among the 29 patients included in our study. A prospective collection and assessment of sequential strains is needed to better understand the dynamic changes occurring in each strain and its relationship to hyperammonemia. Without a control group in our patients, it is difficult to conclusively attribute hyperammonemia to cryptococcal ammonia production. However, we were able to identify risk factors associated with its occurrence and verify ammonia production rates in three of the strains, which strengthen the argument for future studies. Ammonia is known to mediate direct toxicity on many tissues, including kidney cells (32), and is also excreted in the urine. Whether hyperammonemia in the setting of renal failure reflects direct kidney toxicity, reduced excretion, or both, is uncertain. Nevertheless, our observations suggest the possibility that renal failure follows ammonia damage to the kidney as a venue for future study and a new consideration in understanding the pathogenesis of cryptococcosis.

## Conclusion

The occurrence of hyperammonemia was strongly associated with renal failure in the host with a disseminated cryptococcal infection. However, the occurrence of profound and clinically significant hyperammonemia in one patient with renal failure was associated with unusually high levels of urease secretion unique to that cryptococcal strain. Further prospective studies are needed to understand the true incidence, clinical significance, as well as the host-driven and pathogen-driven risk factors for hyperammonemia. Cryptococcal ammonia production can be disrupted *in vitro* using a urease inhibitor, which may have important therapeutic implications and requires further study.

## ACKNOWLEDGMENTS

We would like to acknowledge the valuable contributions of Hernando Gomez, MD, and Dr. Ofer Sadan, MD, in providing input on the study methodology. Their expertise and guidance were instrumental in shaping the research project. We would also like to express our gratitude to the staff of the Investigation Clinical Microbiological Core (ICMC), particularly Sarah Satola, PhD, Ahmad Babiker, MD, and D'Ante Reshaun Gooden, B.S., for their exceptional support in providing microbiological assistance and technical expertise. Their contributions were crucial for the successful completion of this study. We are also grateful to Anne Jedlicka at the Johns Hopkins Bloomberg School of Public Health Genomic Analysis and Sequencing Facility, who provided expert guidance and support for real-time PCR analyses.

This study was a clinical and laboratory collaboration. R.P.B.'s contribution was focused on laboratory aspect of the study while M.S.'s contribution was focused on clinical aspect of the study. R.P.B. performed laboratory analysis, wrote the laboratory methods and

results, and contributed to the discussion. M.S. submitted IRB application, collected the retrospective clinical data, and wrote a significant portion of the manuscript. S.P. contributed to writing and editing sections in the manuscript. S.K. provided guidance in the design of the study and expert infectious disease input to the manuscript, facilitated access to biorepository of specimens, and edited the manuscript. M.F. performed the statistical analysis for clinical (Part 1) of the manuscript. A.C. provided study design guidance for the laboratory analysis section of the manuscript and contributed to editing multiple versions of the manuscript. P.A.K. designed the clinical study, wrote portions of the manuscript, and edited multiple versions of the manuscript.

## AUTHOR AFFILIATIONS

[1]Department of Microbiology and Immunology, Johns Hopkins School of Public Health, Baltimore, Maryland, USA

[2]Division of Neurocritical Care, Department of Neurology, Emory University, Atlanta, Georgia, USA

[3]Division of Neurocritical Care, Department of Neurology, University of Nebraska, Omaha, Nebraska, USA

[4]Division of Infectious Diseases, Department of Medicine, Emory University, Atlanta, Georgia, USA

[5]Office of Vital Statistics, Tennessee Department of Health, Nashville, Tennessee, USA

## AUTHOR ORCIDs

Rosanna P. Baker ⓘ http://orcid.org/0000-0002-9970-6278
Arturo Casadevall ⓘ http://orcid.org/0000-0002-9402-9167
Prem A. Kandiah ⓘ http://orcid.org/0000-0002-7964-0793

## ADDITIONAL FILES

The following material is available online.

### Supplemental Material

**Supplemental material (Spectrum03902-23-s0001.docx).** Table S1; Fig. S1 to S6.

### Open Peer Review

**PEER REVIEW HISTORY (review-history.pdf).** An accounting of the reviewer comments and feedback.

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
