## [Reviewer comments · Microbiology Spectrum]

Microbiology Spectrum

Title: Host and fungal factors both contribute to Cryptococcosis-associated hyperammonemia (Cryptammonia)

Rosanna Baker, Maria Schacter, Steven Phillips, Sheetal Kandiah, Mirza Farrque, Arturo Casadevall, and Prem Kandiah

Corresponding Author(s): Prem Kandiah, Emory University Hospital

Review Timeline:

Submission Date:	November 22, 2023
Editorial Decision:	January 2, 2024
Revision Received:	April 5, 2024
Accepted:	April 9, 2024

Editor: Kirsten Nielsen

Reviewer(s): Disclosure of reviewer identity is with reference to reviewer comments included in decision letter(s). The following individuals involved in review of your submission have agreed to reveal their identity: Derek Fleming (Reviewer #2)

Transaction Report:

DOI: <https://doi.org/10.1128/spectrum.03902-23>

Re: Spectrum03902-23 (Title: Host and fungal factors both contribute to Cryptococcosis-associated hyperammonemia (Cryptammonia))

Dear Dr. Prem Kandiah:

Thank you for the privilege of reviewing your work. Below you will find my comments, instructions from the Spectrum editorial office, and the reviewer comments.

Both reviewers found this to be an interesting study and had only minor criticisms. The authors should address these criticisms in the revised manuscript.

Revision Guidelines

Sincerely,
Kirsten Nielsen
Editor
Microbiology Spectrum

Reviewer #1 (Comments for the Author):

The paper by Baker et al. evaluated the association of hyperammonemia, kidney failure and cryptococcosis. They found that patients with renal failure and cryptococcosis had a much higher risk of hyperammonemia, particularly those patients with disseminated cryptococcosis (presence of cryptococcal cells in the bloodstream). Although small, this retrospective study is well

conducted and well analyzed. Statistic is appropriate. The results are well illustrated, well presented, and well discussed. The methods are clearly outlined. Few issues are below.

1. It is unclear whether the isolates from the 29 patients were *Cryptococcus neoformans* or *Cryptococcus gattii*.
2. It is unclear why only few clinical cryptococcal isolates were characterized for urease activity and other fungal characteristics. After all, the authors should have access to all isolates from the 29 patients (either from blood or/and CSF).

Reviewer #2 (Comments for the Author):

In this study, the authors investigated the risk factors for hyperammonemia in patients with *Cryptococcus neoformans* and *Cryptococcus gattii* infections. A retrospective analysis of 29 patients revealed that 37.9% had hyperammonemia, with renal failure showing a significant association, increasing the risk four-fold. While disseminated cryptococcal infections had a higher risk of hyperammonemia compared to isolated meningitis, a unique strain with increased urease activity was identified in one patient with extreme hyperammonemia. Urease inhibition with an FDA-approved inhibitor was able to dampen urease activity and proliferation by all strains. This manuscript is well written and the experiments have been thoroughly planned and executed. The subject matter sheds light on an underappreciated complication that should garner strong general interest for the spectrum reader base. The most major limitation, being a low sample size, has been addressed. I only have minor concerns and suggestions, listed below, that should be addressed in a revised document.

Minor concerns:

- 1) Growth rate comparisons (i.e. lines 255-263): How does increased culture alkalinity due to ammonia production affect growth? How can you be certain that A1 grows slower in vivo, not just in vitro, especially considering the relatively high starting inoculum for the curves (10^6)?
- 2) Lines 329-330: As mentioned later in the manuscript, it can't be said that these are isolates of the same strain.
- 3) Lines 345-348: It would be worth mentioning that increased urease production has also been found to be associated with longer intracellular persistence (<https://doi.org/10.1371/journal.ppat.1007144>)
- 4) Lines 361-364: This statement would be best positioned immediately following line 336.
- 5) Lines 376-380: It would be worth mentioning that urease inhibition has also been found to be effective in mice and sheep in the context of *Ureaplasma* infection.
- 6) Figure 1: Was a scatter of NH₃ vs Urea also performed?

Suggestions:

- 1) M13 Molecular Typing: It would be very valuable to sequence the urease gene of the different isolates for comparison (eg SNP genotyping), especially for the patient A isolates. Combining with qRT-PCR of Urease gene products would also be interesting. Could lend some weights to the mutability statements by showing variability in the urease itself and/or its production.

Response to Reviewers

Reviewer #1 (Comments for the Author):

The paper by Baker et al. evaluated the association of hyperammonemia, kidney failure and cryptococcosis. They found that patients with renal failure and cryptococcosis had a much higher risk of hyperammonemia, particularly those patients with disseminated cryptococcosis (presence of cryptococcal cells in the bloodstream). Although small, this retrospective study is well conducted and well analyzed. Statistic is appropriate. The results are well illustrated, well presented, and well discussed. The methods are clearly outlined. Few issues are below.

1. It is unclear whether the isolates from the 29 patients were *Cryptococcus neoformans* or *Cryptococcus gattii*.

Thank you for noting this oversight. Only the isolate from one of the 29 patients was *Cryptococcus gattii*. This one patient had an isolated CNS infection (blood culture negative), a normal peak plasma ammonia level of 33 μ mol/L and an acute kidney injury (AKI) stage zero . A positive *Cryptococcus gattii* infection was an inclusion criteria in our study design because it is also a urease producer.

<https://doi.org/10.1111/febs.13229>. We have included the following statement in the methods section:

“Our medical record review revealed 29 patients with cryptococcal infection, comprising 28 cases of *Cryptococcus neoformans* and 1 case of *Cryptococcus gattii*.”

We have also edited the Figure S5 to reflect this breakdown.

2. It is unclear why only few clinical cryptococcal isolates were characterized for urease activity and other fungal characteristics. After all, the authors should have access to all isolates from the 29 patients (either from blood or/and CSF).

Thank you for the important question. *Cryptococcus* infections between 2014 and 2022 were included in our study. However, the Emory ICMC biorepository was founded in 2016 and only began collecting blood stream isolates in 2018. Additionally, the repository focuses on bacterial blood stream isolates (not CSF) and is not geared toward systemic collection of *cryptococcus*. When yeast is found on work up of blood cultures it is usually transferred over to the mycology section of the lab. On occasion, blood culture plate may grow yeast which is sent to ICMC though an established workflow and may account for the availability of isolates from 3 patients in the study. Hence our retrospective collection of *cryptococcus* would not have included all *cryptococcus* isolates (even those from blood).

The methods sections has been edited to make this more clear:

“The Emory Investigational Clinical Microbiological Core (ICMC) biorepository which is an IRB approved biorepository (STUDY00093057) was queried for cryptococcal isolates from patients included in our study. Of the 29 patients, Cryptococcal cells from 3 patients were available and recoverable for analysis and therefore representing a convenience sample of available isolates.”

Reviewer #2 (Comments for the Author):

In this study, the authors investigated the risk factors for hyperammonemia in patients with *Cryptococcus neoformans* and *gattii* infections. A retrospective analysis of 29 patients revealed that 37.9% had hyperammonemia, with renal failure showing a significant association, increasing the risk four-fold. While disseminated cryptococcal infections had a higher risk of hyperammonemia compared to isolated meningitis, a unique strain with increased urease activity was identified in one patient with extreme hyperammonemia. Urease inhibition with an FDA-approved inhibitor was able to dampen urease activity and proliferation by all strains. This manuscript is well written and the experiments have been thoroughly planned and executed. The subject matter sheds light on an underappreciated complication that should garner strong general interest for the spectrum reader base. The most major limitation, being a low sample size, has been addressed. I only have minor concerns and suggestions, listed below, that should be addressed in a revised document.

Minor concerns:

1) Growth rate comparisons (i.e. lines 255-263): How does increased culture alkalinity due to ammonia production affect growth? How can you be certain that A1 grows slower in vivo, not just in vitro, especially considering the relatively high starting inoculum for the curves (10^6)?

Increased culture alkalinity due to ammonia production was shown to inhibit growth of *C. neoformans* at pH values of 8.5 or higher (Baker and Casadevall, 2023) but in the experiments conducted in the current study, the pH of cultures in the presence or absence of AHA remained at or below neutral pH. We cannot be certain that the A1 isolate would also grow more slowly in vivo, since previous studies showed slower growth of urease-deficient compared to urease-positive *C. neoformans* in vitro but faster replication for the urease deletion strain inside macrophages (Fu et al., 2018).

The more pertinent observation in this study is that growth of WT and all patient isolates were inhibited to comparably low levels in the presence of AHA as this is expected to prove helpful in the clinical setting. The following statement has been added to the results:

“The growth inhibitory effect of AHA is consistent with previous reports of decreased in vitro growth rates for urease-deficient *C. neoformans*”

2) Lines 329-330: As mentioned later in the manuscript, it can't be said that these are isolates of the same strain.

We thank the reviewer for suggesting this clarification. The sentence has been edited to read as follows: “Interestingly, subsequent isolates from this patient obtained on days 3, 5 and 12 on antifungal therapy revealed a reduction in urease activity.”

3) Lines 345-348: It would be worth mentioning that increased urease production has also been found to be associated with longer intracellular persistence (<https://doi.org/10.1371/journal.ppat.1007144>)

Thank you for the insightful suggestion. We have edited the sentence to read:

“For instance, faster growth rates are associated with increased incidence of lytic release from phagocytes while slower intracellular growth rates, that have been correlated with increased urease

activity, prolong the persistence of cryptococcal cells within the phagosome.”

4) Lines 361-364: This statement would be best positioned immediately following line 336.

We are grateful for the suggestion and have edited the fourth paragraph of the discussion to incorporate the statement originally from Lines 361-364.

5) Lines 376-380: It would be worth mentioning that urease inhibition has also been found to be effective in mice and sheep in the context of *Ureaplasma* infection.

Thank you for this suggestion. We added the following statement in line 380 to 383: “In the context of *Ureaplasma* infections, the use of a urease inhibitor has been demonstrated to limit the growth of *Ureaplasma* species in vitro and in vivo in mice and sheep(25, 26), and to resolve *Ureaplasma*-induced hyperammonemia in mice(27).”

6) Figure 1: Was a scatter of NH3 vs Urea also performed?

Thank you for this suggestion. We chose to focus on creatinine to correlate renal dysfunction as a risk factor for hyperammonemia because BUN would be a substrate for the urease enzyme and may not reflect the level of renal failure. We have provided below the requested NH3 vs. BUN scatter plot. There is the possibility that a low urea: creatinine ratio may predict a urease producing infection however that was not clearly reflected in our data. This may be confounded by multiple factors: 1. Rate of renal failure may outpace urease activity 2. Variability in urease activity with each strain 3. Nutritional state of the patient as BUN may be elevated in a catabolic state or anabolic resistance during sepsis. 4. The small sample size of our study especially pertaining to patient with moderate to severe hyperammonemia (i.e. > 100 $\mu\text{mol/L}$). We have added the NH3 vs. Urea scatter plot as supplementary data labeled : Figure S6.

Figure-S6: Scatter plot with a linear fitted line showing the relationship between admission Blood Urea Nitrogen and peak NH3, Cryptammonia study, Emory data warehouse, 2013-2022 (N=28)

Suggestions:

1) M13 Molecular Typing: It would be very valuable to sequence the urease gene of the different isolates for comparison (eg SNP genotyping), especially for the patient A isolates. Combining with qRT-PCR of Urease gene products would also be interesting. Could lend some weights to the mutability statements by showing variability in the urease itself and/or its production.

We thank the reviewer for these insightful suggestions. We have completed DNA sequence analysis of the urease open reading frame and quantitative real-time PCR to compare urease expression in WT and the patient isolates. The sequence data have been added to supplementary Figure S1 and the expression analyses have been added as Figure 5 in the revised manuscript. This analysis has improved the manuscript as it revealed that increased ammonia production by the hyperammonemia patient isolates was not due to overproduction of urease. We also extended our analysis to compare intracellular urease activity which, in agreement with the expression data, showed lower activity for patient isolate A1 compared to WT. These observations suggest that increased ammonia production by the hyperammonemia patient isolate is due to increased extracellular transport of urease.

Re: Spectrum03902-23R1 (Title: Host and fungal factors both contribute to Cryptococcosis-associated hyperammonemia (Cryptammonia))

Dear Dr. Prem Kandiah:

Reviewers determined the manuscript was acceptable

Your manuscript has been accepted, and I am forwarding it to the ASM production staff for publication. Your paper will first be checked to make sure all elements meet the technical requirements. ASM staff will contact you if anything needs to be revised before copyediting and production can begin. Otherwise, you will be notified when your proofs are ready to be viewed.

Sincerely,
Kirsten Nielsen
Editor
Microbiology Spectrum

Reviewer #1 (Comments for the Author):

The authors responded to my comments in a satisfactory manner.

Reviewer #2 (Comments for the Author):

I have no further comments.